# Assessment of Evolutionary Relationships for Prioritization of Myxobacteria for Natural Product Discovery

**DOI:** 10.3390/microorganisms9071376

**Published:** 2021-06-24

**Authors:** Andrew Ahearne, Hanan Albataineh, Scot E. Dowd, D. Cole Stevens

**Affiliations:** 1Department of BioMolecular Sciences, School of Pharmacy, University of Mississippi, Oxford, MS 38677, USA; aahearne@go.olemiss.edu (A.A.); haalbata@go.olemiss.edu (H.A.); 2MR DNA, Molecular Research LP, Shallowater, TX 79363, USA; sdowd@mrdnalab.com

**Keywords:** myxobacteria, *Myxococcus* sp., *Corallococcus* sp., *Melittangium* sp., *Archangium* sp., biosynthetic gene clusters

## Abstract

Discoveries of novel myxobacteria have started to unveil the potentially vast phylogenetic diversity within the family Myxococcaceae and have brought about an updated approach to myxobacterial classification. While traditional approaches focused on morphology, 16S gene sequences, and biochemistry, modern methods including comparative genomics have provided a more thorough assessment of myxobacterial taxonomy. Herein, we utilize long-read genome sequencing for two myxobacteria previously classified as *Archangium primigenium* and *Chondrococcus macrosporus*, as well as four environmental myxobacteria newly isolated for this study. Average nucleotide identity and digital DNA–DNA hybridization scores from comparative genomics suggest previously classified as *A. primigenium* to instead be a novel member of the genus *Melittangium*, *C. macrosporus* to be a potentially novel member of the genus *Corallococcus* with high similarity to *Corallococcus exercitus*, and the four isolated myxobacteria to include another novel *Corallococcus* species, a novel *Pyxidicoccus* species, a strain of *Corallococcus exiguus*, and a potentially novel *Myxococcus* species with high similarity to *Myxococcus stipitatus*. We assess the biosynthetic potential of each sequenced myxobacterium and suggest that genus-level conservation of biosynthetic pathways support our preliminary taxonomic assignment. Altogether, we suggest that long-read genome sequencing benefits the classification of myxobacteria and improves determination of biosynthetic potential for prioritization of natural product discovery.

## 1. Introduction

Over the last decade, 34 novel species of myxobacteria have been described including representatives from 10 newly described genera within the order *Myxococcales* (Appendix A) [1,2,3,4,5,6,7,8,9,10,11,12,13,14]. Prevalent in soils and marine sediments, predatory and cellulolytic myxobacteria contribute to nutrient cycling within microbial food webs. Perhaps most-studied for their cooperative lifestyles, myxobacteria have been an excellent resource for investigations concerning developmental multicellularity and two-component signaling, swarming motilities and predatory features, and the discovery of biologically active metabolites [15,16,17,18,19,20,21,22,23]. Each of these areas of interest have benefited from the increased utility and accessibility of next-generation sequencing (NGS) technologies. The driving force behind the recent surge in efforts to discover novel species of myxobacteria can also be attributed to advances in sequencing technologies. Genome sequencing of myxobacteria has demonstrated that they possess large genomes replete with biosynthetic gene clusters, and myxobacteria have recently been deemed a “gifted” taxon for the production of specialized metabolites with drug-like properties [24,25,26,27,28,29]. These efforts, combined with a thorough metabolic survey of over 2000 strains within the order *Myxococcales*, concluded that the odds of novel metabolite discovery increase when exploring novel genera of myxobacteria [30]. Motivated by these observations, we sought to isolate novel myxobacteria from lesser-studied North American soils.

Recently, comparative genomic analyses have been utilized to provide efficient preliminary classification of novel myxobacteria, and we considered that such an approach would expedite prioritization of strains for future metabolic studies [3,8,11,31,32,33,34,35,36,37]. While traditional myxobacterial classification efforts relied on morphology, biochemistry, and the conservation of 16S gene sequences, updated methods including genome-based taxonomy have provided excellent preliminary taxonomic classification of myxobacterial isolates [38,39,40]. Considering that genome sequencing would also afford the biosynthetic potential of any isolated myxobacteria, we sought to employ long-read sequencing to generate high-quality draft genomes hoping to avoid fragmented, partial biosynthetic pathways. For example, of the 11 currently sequenced myxobacteria from the genus *Corallococcus*, 68% of the 621 total putative biosynthetic gene clusters (BGCs) predicted by the analysis platform AntiSMASH are positioned on a contig edge and are potentially incomplete (Appendix A). In fact, the only two *Corallococcus* genomes sequenced with long-read techniques (*Corallococcus coralloides* DSM 2259^T^ and *C. coralloides* strain B035) each included 34 predicted BGCs with none located on a contig edge [41,42]. Ideally, larger contigs generated from long-read sequencing might benefit the comparative genomic analyses and provide a more complete assessment of biosynthetic potential.

In addition to four environmental isolates of putative myxobacteria included in this study, we acquired two previously characterized myxobacteria from the American Type Culture Collection (ATCC): *Archangium primigenium* ATCC 29,037 and *Chondrococcus macrosporus* ATCC 29039. Previously miscategorized as *Polyangium primigenium*, the original morphological descriptions for *A. primigenium* were remarkably apt for the strain acquired from the ATCC and cultivated in our lab, including obvious fruiting body formation and carotene-like pigmentation (Figure 1) [43,44]. The original description of *A. primigenium* fruiting bodies initially piqued our interest in the strain as members of the genus *Archangium* typically do not or very rarely form defined fruiting bodies when cultivated with standard laboratory conditions [45,46]. *Archangium* species have previously been referred to as “degenerate forms’’ of myxobacteria due to diminished fruiting bodies with no sporangioles or absent fruiting body formation [46]. Comparatively, little historical data is available for *C. macrosporus* ATCC 29039. The strain was deposited at the ATCC by distinguished taxonomist Professor V. B. D. Skerman and was subsequently included in a methodology study focused on isolating myxobacteria from soils [47,48,49]. The decision to change the genus *Chondrococcus* to instead be *Corallococcus* has been validated with many novel *Corallococcus* species being described afterwards [8,40,50]. However, we were curious to determine the status of *C. macrosporus* ATCC 29039. Considering the proposed reassignment of *Corallococcus macrosporus* DSM 14697^T^ to the genus *Myxococcus*, it was unclear if *C. macrosporus* ATCC 29,039 should also be reassigned. Both characterized using traditional approaches that heavily relied on morphology, we sought to determine how genomic comparisons might impact the current taxonomic assignments of these available myxobacteria.

## 2. Materials and Methods

### 2.1. Bacterial Strains and Growth Conditions

*A. primigenium* and *C. macrosporus* were procured from the ATCC as strain numbers ATCC 29037 and ATCC 29039, respectively. The remaining strains were isolated from soil as described later. All strains were cultured either on VY/2 or VY/4 agar plates (5 or 2.5 g/L baker’s yeast, 1.5 g/L CaCl_2_·2H_2_O, 0.5 mg/L vitamin B12, 15 g/L agar, pH 7.2). Swarming and fruiting bodies on agar plates were observed under a Zeiss discovery V12 stereo microscope and photographed using a Zeiss axiocam105.

### 2.2. Isolation of Environmental Myxobacteria

Soil samples, collected in Asheville, NC and Tryon, SC, were taken from the base of trees and dried in open air before storage. Detailed location data are provided as Appendix A. Myxobacteria were isolated using a slightly modified Coli-spot method [51]. A 1 mg/mL solution of cycloheximide/nystatin was used to wet the soil sample to a paste-like consistency before inoculation onto an *Escherichia coli* baited WAT agar plate (1 g/L CaCl_2_·2H_2_O, 15 g/L agar, 20 mM HEPES). To prepare the baiting plate, a lawn of *E. coli* was grown overnight on tryptone soya broth (TSB) with agar (1.5%), and the cells were scraped and suspended in 2 mL of sterile deionized water. Four hundred μL of the *E. coli* suspension was spread over the surface of a WAT agar plate to create a bait circle of approximately 2 inches in diameter and let dry. Once the *E. coli* was dried, a pea sized amount of soil paste was placed at the center of the bait circle. Plates were incubated at 25 °C for up to a month, and degradation of the *E. coli* was monitored over time. Visible degrading swarms were seen after a few days, and swarm edges or fruiting bodies were passaged onto VY/4 media for purification. Purification was accomplished by repeated swarm edge transfer.

### 2.3. Genomic DNA Isolation, Sequencing, Assembly, and Annotation

Genomic DNA for NGS was obtained from actively growing bacteria on VY/2 or VY/4 plates using NucleoBond high molecular weight DNA kit (Macherey-Nagel, Bethlehem, PA, USA). The quantity and quality of the extraction were checked by Nanodrop (Thermo Scientific NanoDrop One) and followed by Qubit quantification using Qubit^®®^ dsDNA HS Assay Kit (ThermoFisher Scientific, Suwanee, GA, USA).

Sequencing for all samples was performed on a Pacific Biosciences single-molecule real-time (SMRT) sequencing platform at the MR DNA facility (Shallowater, TX, USA). The SMRTbell libraries for the sample were prepared using the SMRTbell Express Template Prep Kit 2.0 (Pacific Biosciences, Menlo Park, CA, USA) following the manufacturer’s user guide. Following library preparation, the final concentration of each library was measured using the Qubit^®^ dsDNA HS Assay Kit (ThermoFisher Scientific, Suwanee, GA, USA), and the average library sizes were determined using the Agilent 2100 Bioanalyzer (Agilent Technologies, Santa Clara, CA, USA). Each library pool was then sequenced using the 10-h movie time on the PacBio Sequel (Pacific Biosciences, Menlo Park, CA, USA). De Novo Assembly of each genome was accomplished using the PacBio SMRT Analysis Hierarchical Genome Assembly Process (HGAP). Genome annotation was done using Rapid Annotation using Subsystem Technology (RAST) with further annotation requested by the NCBI Prokaryotic Genome Annotation Pipeline [52]. Sequencing data have been deposited in NCBI under the accession numbers JADWYI000000000.1, JAFIMU000000000, JAFIMS000000000, JAFIMT000000000, CP071090, and CP071091 for strains *A. primigenium*, *C. macrosporus*, NCSPR001, NCCRE002, SCPEA002, and SCHIC003, respectively.

### 2.4. Comparative Genomic Studies

The genome sequence data were uploaded to the Type (Strain) Genome Server (TYGS), a free bioinformatics platform available under https://tygs.dsmz.de (accessed 10 January 2021), for a whole genome-based taxonomic analysis. TYGS was used to calculate the dDDH values and construct minimum evolution trees using the Genome BLAST Distance Phylogeny approach (GBDP) [53,54]. GBDP trees were visualized using MEGA-X [55]. The average nucleotide identity (ANI) was calculated using the ANI/AAI-Matrix calculator [56,57].

### 2.5. BIG-SCAPE Analysis

Genome data for all myxobacteria belonging to the *Cystobacterineae* suborder were downloaded from the NCBI database. A list of all myxobacteria used in this analysis are listed in Appendix A. These genomes in addition to genomes of *A. primigenium*, *C. macrosporus*, and the environmental isolates were analyzed by the AntiSMASH platform (version 5 available at https://docs.antismash.secondarymetabolites.org; accessed 1 February 2021) to assess specialized metabolite gene clusters using the “relaxed” strictness setting [58,59]. A total of 1826 predicted BGCs (.gbk files) were then processed locally using the BiG-SCAPE program (version 20181005, available at https://git.wageningenur.nl/medema-group/BiG-SCAPE; accessed 1 February 2021), with the MiBIG database (version 2.0 available at https://mibig.secondarymetabolites.org; accessed 1 February 2021) as reference [60,61]. BiG-SCAPE analysis was supplemented with Pfam database version 33.1 [62]. The singleton parameter in BiG-SCAPE was selected to ensure that BGCs with distances lower than the default cutoff distance of 0.3 were included in the corresponding output data. The hybrids-off parameter was selected to prevent hybrid BGC redundancy. Generated network files separated by BiG-SCAPE class were combined for visualization using Cytoscape version 3.8.2 (http://www.cytoscape.org; accessed 1 February 2021) [63]. Annotations associated with each BGC were included in Cytoscape networks by importing curated tables generated by BiG-SCAPE.

## 3. Results

### 3.1. Comparative Genomics and Taxonomic Assessment of Archangium Primigenium, Chondrococcus Macrosporus, and Environmental Isolates

Genome sequencing provided high quality draft genomes for each of the six investigated myxobacteria, as indicated by the summary of general features in Table 1. The total genome sizes ranged from ~9.5–13 Mb, and the %GC content varied around ~69–71%. Of the six genomes, both environmental strains SCHIC003 and SCPEA002 were assembled on a single contig. Overall, the assemblies for each genome provided much lower total contig counts (1–17) than recently sequenced myxobacterial genomes [3,8]. Interestingly, a minimum evolution of phylogenetic trees generated from the whole genome sequence data clustered *A. primigenium* with *Melittangium boletus* DSM 14713^T^ and not with the three currently sequenced strains from the genus *Archangium* (Figure 2, Appendix A). Accordingly, ANI and dDDH values supported the placement of *A. primigenium* in the genus *Melittangium* (Table 2) as a novel species with both values well below the established cutoffs for classification of distinct species (<95% ANI; <70% dDDH) [31,34,35,37,64]. These data suggest *A. primigenium* is currently misclassified as a member of the genus *Archangium* and should instead be placed in the genus *Melittangium*.

The calculated ANI and dDDH values for the sequenced *C. marcosporus* strain acquired from the ATCC support the original assignment to the genus *Chondrococcus*, now *Corallococcus* [31,50]. As opposed to the recently reclassified *Myxococcus macrosporus* DSM 14697^T^, previously *Corallococcus macrosporus*, the minimum evolution phylogenetic tree suggested *C. macrosporus* ATCC 29039 to be a member or the genus *Corallococcus* most similar to *Corallococcus exercitus* DSM 108849^T^ (Figure 2, Appendix A) [50]. The isolated strains NCCRE002 and NCSPR001 were also determined to be members of the genus *Corallococcus* (Figure 2, Appendix A). Comparative genome analyses implied that strain NCCRE002 is an isolate of *Corallococcus exiguus* DSM 14696^T^. However, the ANI and GBDP trees suggested that strain NCSPR001 is a novel member of the genus *Corallococcus* most similar to *Corallococcus coralloides* DSM 2259^T^ (Table 3).

The isolated SCHIC003 and SCPEA002 strains were initially determined to be members of the genus *Myxococcus*. However, inclusion of sequenced representatives from the genus *Pyxidicoccus* (considered to be synonymous with *Myxococcus*) [3] in our comparative analysis grouped strain SCPEA002 within the *Pyxidicoccus* clade (Figure 2, Appendix A). Most similar to *Pyxidicoccus caerfyrddinensis* CA032A^T^, dDDH and ANI analysis suggested the SCPEA002 strain to be a novel member of the genus *Pyxidicoccus* (Table 4). Similarly, comparative genome analysis determined that strain SCHIC003 is likely be a novel member of the genus *Myxococcus*, albeit highly similar to *Myxococcus stipitatus* DSM 14675^T^ with ANI and dDDH values just below the cutoffs for species differentiation [31,37,64] (Table 4 and Figure 2, Appendix A).

### 3.2. Biosynthetic Potential and Genus Level Correlations

Analysis of our draft genomes using the biosynthetic pathway prediction platform AntiSMASH revealed a range of 29–42 total predicted BGCs with *C. macrosporus* including the highest total of BGCs. However, the draft genome for *C. macrosporus* also included the highest total of four partial BGCs positioned on the edges of contigs. No BGCs occurring on contig edges were observed from *A. primigenium*, NCSPR001, or SCPEA002. All of the sequenced strains included highly similar (≥75% similarity score) biosynthetic pathways for the signaling terpene geosmin [65,66], the signaling lipids VEPE/AEPE/TG-1 [67,68], and carotenoids [69,70,71,72] (Figure 3). Excluding SCHIC003, each genome included a BGC highly homologous to the pathway associated with the myxobacterial siderophore myxochelin [73,74]. Pathways somewhat similar (similarity scores of 66%) to the myxoprincomide-c506 BGC were observed in every genome except the *A. primigenium* genome [75]. Clusters with ≥75% similarity to pathways from *M. stipitatus* DSM 14675^T^ associated with the metabolites rhizopodin [76,77] and phenalamide A2 [78] were observed in the SCHIC003 draft genome as well as clusters also present in the *M. stipitatus* DSM 14675^T^ genome deposited in the AntiSMASH database [79], including the dkxanthene [80], fulvuthiacene [81], and violacein [82,83,84] BGCs (Figure 4). Considering previously characterized BGCs from each genus associated with the six investigated myxobacteria, the corallopyronin BGC from *C. coralloides* B035 [85,86] was absent from all three of the putative *Corallococcus* strains, the melithiazol BGC from *Melittangium lichenicola* Me I46 [87] was not present in *A. primigenium*, and neither the disciformycin/gulmirecin BGC [88,89] or the pyxidicycline BGC [90] from *Pyxidicoccus fallax* were present in SCPEA002.

Utilizing the BiG-SCAPE platform to render BGC sequence similarity networks, we sought to determine the extent of homology between BGCs from our six sequenced myxobacteria and BGCs from all currently sequenced members of the suborder *Cystobacterineae* [91]. The resulting sequence similarity network included 1080 BGCs connected by 3046 edges (not including self-looped nodes/singletons) and depicted genus-level homologies across all BGCs from the newly sequenced myxobacteria corroborating our suggested taxonomic assignments (Figure 5 and Table 5). For example, BGCs from the three newly sequenced samples *C. macrosporus*, NCSPR001, and NCCRE002 were almost exclusively clustered with BGCs from members of the genus *Corallococcus*, and BGCs from SCHIC003 and SCPEA002 samples clustered with the genera *Myxococcus* and *Pyxidicoccus* (Figure 5). However, SCPEA002 BGCs do not cluster as frequently with *Pyxidicoccus* BGCs as they do *Myxococcus* BGCs, and the majority (76.5%) were not clustered with any BGC within the network (Table 5). This is likely due to the highly fragmented nature of available *Pyxidicoccus* genomes resulting in many incomplete or partial BGCs. Therefore, few *Pyxidicoccus* pathways appear in the similarity network, and the percentage of unique pathways associated with SCPEA002 is likely overestimated. Regardless, the limited number of SCPEA002 BGCs clustered with BGCs from *Myxococcus/Pyxidicoccus* genomes indicates a potential to discover novel metabolites despite placement in the highly scrutinized clade. The only clustered groups with numerous edges formed between BGCs from the genera *Myxococcus* and *Corallococcus* included characterized biosynthetic pathways for ubiquitous signaling lipids VEPE/AEPE/TG-1, carotenoids, and the siderophore myxochelin as well as two uncharacterized BGCs predicted to produce ribosomally synthesized and post-translationally modified peptides (RiPPs).

Interestingly, a total of 23 *A. primigenium* BGCs (out of 32 BGCs) appear as singletons in the network with no homology to any of the included BGCs from *Cystobacterineae*. In fact, aside from the VEPE/AEPE/TG-1 cluster and a terpene cluster that included members of the genera *Archangium* and *Cystobacter*, all remaining BGCs from *A. primigenium* had connecting edges to BGCs from *Melittangium boletus* DSM 14713^T^. Out of 21 edges formed by *A. primigenium* in the network, four edges were formed with four species of *Corallococcus* (a total of 11 *Corallococcus* species in the network), four edges were formed with all species of *Cystobacter* (three species in the network), six edges were formed with all species of *Archangium* (three species in the network), and seven edges were formed with the only *Melittangium* species in the network, *M. boletus* DSM 14713^T^. Overall, these data corroborate our preliminary taxonomic assignments and suggest that the prioritization of *A. primigenium* for subsequent discovery efforts is most likely to yield novel metabolites.

## 4. Discussion

As novel myxobacteria continue to be isolated and explored for natural product discovery, efficient approaches for approximate taxonomic placement will assist the prioritization of lesser studied genera. Utilizing long-read genome sequencing and comparative genomic analyses, we determine preliminary taxonomic placement for four myxobacteria isolated from North American soils and two myxobacteria deposited at the ATCC. This approach indicated that previously classified *A. primigenium* ATCC 29037 is instead a novel member of the genus *Melittangium*, and that three of our four environmental isolates included potentially novel members of the genera *Corallococcus, Myxococcus,* and *Pyxidicoccus*. Previously classified *Chondorococcus macrosporus* ATCC 29039 was also determined to be a potentially novel member of the genus *Corallococcus*, with high similarity to *C. exercitus* DSM 108849^T^ and phylogenetically distinct from *M. macrosporus* DSM 14697^T^ previously assigned to the genus *Corallococcus*. Subsequent bioinformatic analysis of biosynthetic pathways included in the newly sequenced genomes corroborated our preliminary taxonomic placements for each sample. Ultimately, this process identified *A. primigenium* to be a member of the lesser studied genus *Melittangium* and indicated that it should be prioritized for continued natural product discovery efforts. Of the environmental isolates, BGCs from SCPEA002 were determined to include the least amount of overlap with BGCs from other *Myxococcus/Pyxidicoccus* species. While environmental isolates SCHIC002 and NCSPR001 were also identified as novel members of the genera *Myxococcus* and *Corallococcus*, respectively, the apparent overlap in BGCs from thoroughly explored myxobacteria determined from sequence similarity network analysis suggests a limited potential for discovery of novel specialized metabolites. Overall, comparative genomic techniques including the assessment of biosynthetic potential enabled a phylogenetic approximation and suggested prioritization of *A. primigenium* for natural product discovery efforts from a sample set of six newly sequenced myxobacteria.

## Figures and Tables

**Figure 1 microorganisms-09-01376-f001:**
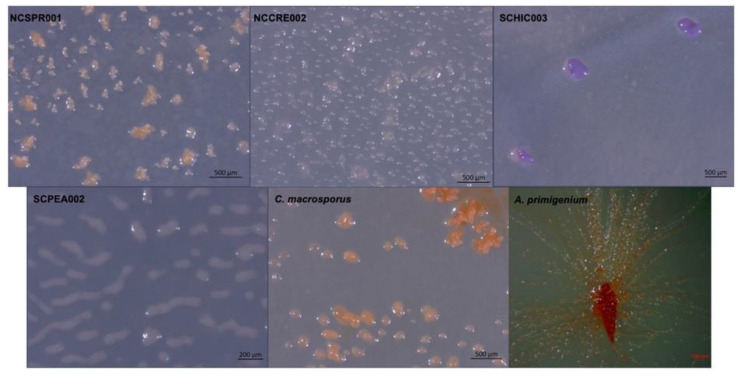
Myxobacterial fruiting bodies from strains NCSPR001, NCCRE002, SCHIC003, SCPEA002, and the strains *C. macrosporus* ATCC 29,039 and *A. primigenium* ATCC 29037.

**Figure 2 microorganisms-09-01376-f002:**
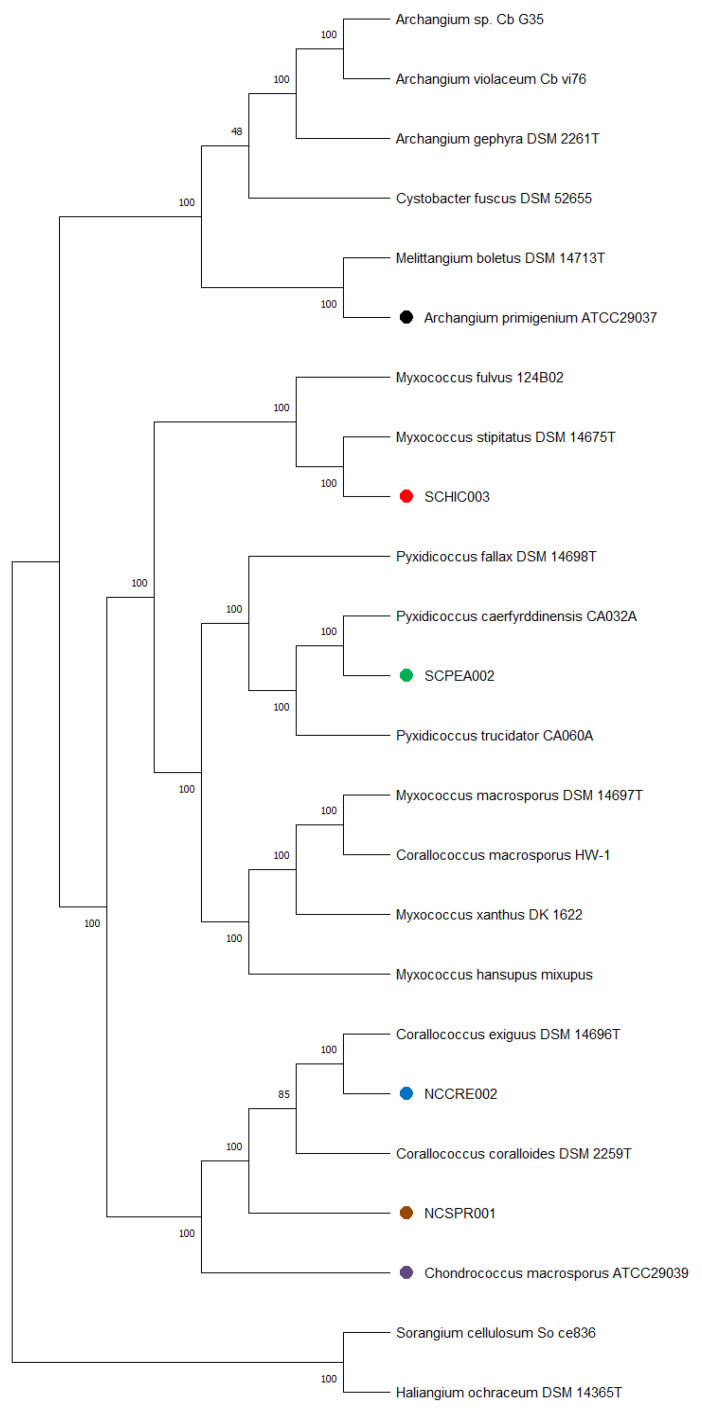
Minimum evolution tree from the whole genomes of different myxobacteria including the six strains under investigation in this study using the GBDP approach. The numbers in bold above branches are GBDP pseudo-bootstrap support values > 60% from 100 replications, with an average branch support of 100.0%. Branch pseudo-bootstraps less than 50% are not shown. The numbers below branches are branch lengths scaled in terms of GBDP distance formula d5. The tree was rooted at the midpoint.

**Figure 3 microorganisms-09-01376-f003:**
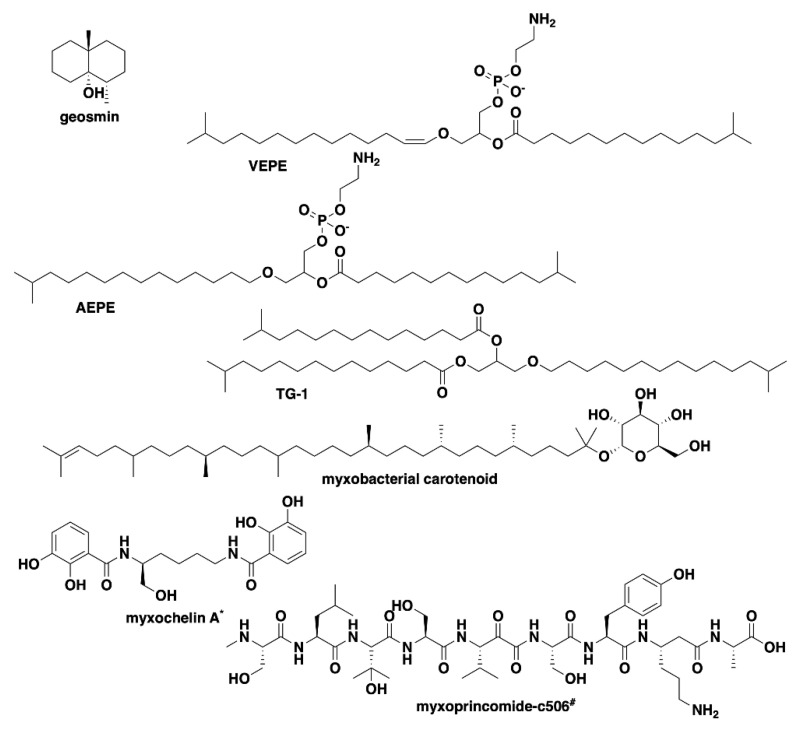
Common specialized metabolites from myxobacteria associated with characterized BGCs present in the six investigated strains of myxobacteria. ^*^ Myxochelin BGC not present in SCHIC003 genome data. ^#^ Myxoprincomide BGC not present in *A. primigenium* genome data.

**Figure 4 microorganisms-09-01376-f004:**
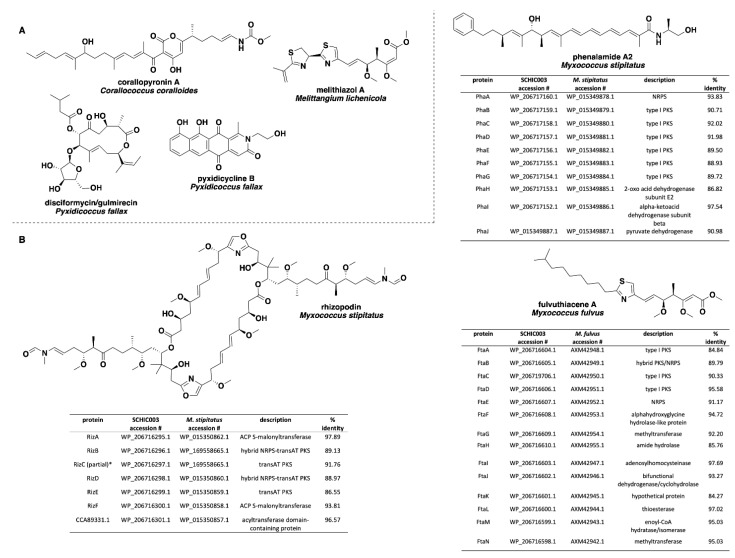
(**A**) Specialized metabolites produced by members of the genera *Corallococcus, Melittangium,* and *Pyxidicoccus* with no associated BGCs observed in any of the six investigated myxobacterial strains. (**B**) Comparisons of the rhizopodin, phenalamide A2, and fulvuthiacene BGCs from SCHIC003 genome data and the characterized pathways from *M. stipitatus* and *M. fulvus.* All SCHIC003 gene products, excluding RizC, had coverages ≥99% with the indicated homolog. ^*^ RizC located on a contig edge and is incomplete in SCHIC003 genome data.

**Figure 5 microorganisms-09-01376-f005:**
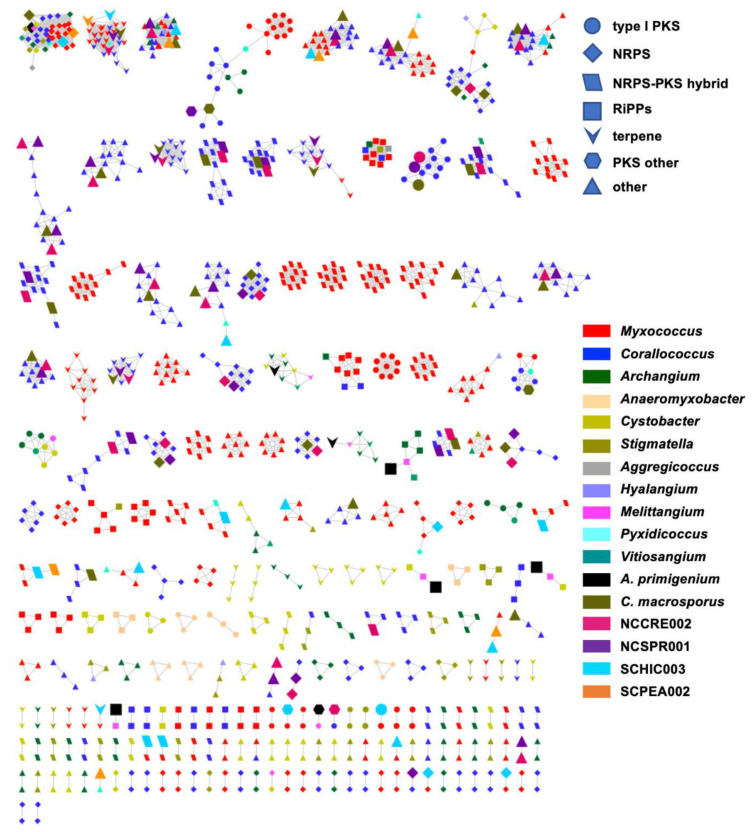
BiG-SCAPE BGC sequence similarity networks (c = 0.3) as visualized with Cytoscape. The network is generated from *A. primigenium*, *C. macrosporus*, NCCRE002, NCSPR001, SCHIC003, SCPEA002, and all myxobacteria belonging to the Cystobacterineae suborder with genomes deposited in NCBI. Each node represents one BGC identified by AntiSMASH 5.0, where the colors and shapes of the nodes represent different genera and AntiSMASH-predicted classes, respectively. Nodes representing BGCs from newly sequenced myxobacteria included in this study are enlarged. BGCs included as singletons in the original BiG-SCAPE analysis removed.

**Table 1 microorganisms-09-01376-t001:** Genome properties and general features of myxobacteria under investigation in this study.

Species	Size (bp)	CDS	GC%	N50	L50	Contigs	Coverage
*A. primigenium*	9,491,554	7873	70.7%	9,468,833	1	3	441x
*C. macrosporus*	9,811,739	7977	70.4%	1,094,727	2	17	300x
NCSPR001	9,785,177	8033	70.1%	9,343,940	1	3	312x
NCCRE002	10,538,407	8589	69.7%	3,024,381	2	8	479x
SCPEA002	13,211,253	10,588	69.6%	N/A	1	1	144x
SCHIC003	10,367,529	8339	68.6%	N/A	1	1	301x

**Table 2 microorganisms-09-01376-t002:** 16S rRNA identity, ANI, and dDDH values for pairwise comparisons between *A. primigenium* with the most similar fully sequenced myxobacteria.

Species	16s rRNA	dDDH	ANI
*M. boletus* DSM 14713^T^	98.89%	29.5	86.1%
*C. fuscus* DSM 52655	98.7%	24.5	83.29%
*A. gephyra* DSM 2261^T^	97.72%	23.2	81.41%
*S. aurantiaca* DW43-1	96.06%	20	78.9%
*M. macrosporus* DSM 14697^T^	96.63%	19.8	78.85%

**Table 3 microorganisms-09-01376-t003:** Differentiation chart comparing *C. macrosporus* ATCC 29039, NCCRE002, and NCSPR001 draft genome data with sequenced members of the genus *Corallococcus*. The top half uses total genome comparison methods (ANI and dDDH) while the bottom half uses 16S rRNA sequence for pairwise comparison. Orange shading represents species that would be designated as the same using the designated method. Blue shading represents unique species using the designated method, <98.65% 16S identity%, or < 95%/70% for ANI/dDDH.

	NCCRE002	NCSPR001	*Chondrococcus macrosporus*	*Corallococcus interemptor* T	*Corallococcus terminator* T	*Corallococcus sicarius* T	*Corallococcus praedator* T	*Corallococcus macrosporus* HW1	*Corallococcus llansteffanensis* T	*Corallococcus exiguus* T	*Corallococcus exercitus* T	*Corallococcus coralloides* T	*Corallococcus carmarthensis* T	*Corallococcus aberystwythensis* T	*Corallococcus* Z5C101001	*Corallococcus* ZKHCc1_1396	*Corallococcus* CA053C
NCCRE002	**100%**	**dDDH: 51**	**41**	**44**	**29**	**29**	**30**	**21**	**30**	**66**	**43**	**54**	**43**	**43**	**34**	**29**	**30**
		**ANI: 94**	**91**	**92**	**86**	**86**	**86**	**81**	**87**	**96**	**91**	**94**	**92**	**91**	**88**	**86**	**86**
NCSPR001	**99.8**	**100%**	**dDDH: 41**	**45**	**29**	**30**	**30**	**21**	**30**	**51**	**43**	**54**	**43**	**43**	**34**	**29**	**30**
			**ANI: 91**	**92**	**86**	**87**	**87**	**81**	**87**	**93**	**92**	**94**	**91**	**91**	**88**	**87**	**87**
***Chondrococcus macrosporus***	**99.15**	**99.22**	**100%**	**dDDH: 42**	**30**	**30**	**30**	**21**	**31**	**41**	**51**	**42**	**43**	**44**	**34**	**30**	**31**
				**ANI: 91**	**86**	**87**	**87**	**81**	**87**	**91**	**94**	**91**	**92**	**92**	**89**	**87**	**87**
***Corallococcus interemptor* T**	**99.8**	**99.87**	**99.35**	**100%**	**dDDH: 29**	**30**	**30**	**21**	**30**	**44**	**42**	**46**	**42**	**42**	**34**	**30**	**30**
					**ANI: 86**	**87**	**87**	**81**	**87**	**92**	**91**	**92**	**91**	**91**	**88**	**87**	**87**
***Corallococcus terminator* T**	**99.03**	**98.96**	**99.09**	**98.96**	**100%**	**dDDH: 35**	**49**	**21**	**35**	**29**	**30**	**29**	**30**	**30**	**31**	**42**	**34**
						**ANI: 89**	**93**	**81**	**89**	**86**	**87**	**86**	**87**	**87**	**87**	**91**	**89**
***Corallococcus sicarius* T**	**98.89**	**98.83**	**98.98**	**98.83**	**99.61**	**100%**	**dDDH: 35**	**21**	**50**	**30**	**31**	**30**	**31**	**31**	**31**	**43**	**50**
							**ANI: 89**	**81**	**93**	**86**	**87**	**87**	**87**	**87**	**87**	**89**	**93**
***Corallococcus praedator* T**	**99.03**	**98.96**	**99.09**	**98.96**	**100**	**99.61**	**100%**	**dDDH: 21**	**36**	**30**	**31**	**30**	**31**	**31**	**32**	**43**	**35**
								**ANI: 81**	**89**	**86**	**87**	**87**	**87**	**87**	**87**	**92**	**89**
***Corallococcus macrosporus* HW1**	**97.73**	**97.66**	**98.37**	**97.79**	**97.72**	**97.72**	**97.72**	**100%**	**dDDH: 22**	**21**	**21**	**21**	**21**	**21**	**21**	**21**	**21**
									**ANI: 81**	**81**	**81**	**81**	**81**	**81**	**81**	**81**	**81**
***Corallococcus llansteffanensis* T**	**98.83**	**98.76**	**98.89**	**98.76**	**99.54**	**99.93**	**99.54**	**97.73**	**100%**	**dDDH: 30**	**32**	**31**	**32**	**31**	**33**	**36**	**54**
										**ANI: 87**	**88**	**87**	**87**	**87**	**88**	**89**	**94**
***Corallococcus exiguus* T**	**99.93**	**99.87**	**99.22**	**99.87**	**99.09**	**98.96**	**99.09**	**97.79**	**98.89**	**100%**	**dDDH: 43**	**54**	**44**	**43**	**34**	**30**	**30**
											**ANI: 91**	**94**	**92**	**91**	**88**	**86**	**86**
***Corallococcus exercitus* T**	**99.02**	**99.09**	**99.87**	**99.22**	**99.22**	**99.09**	**99.22**	**98.37**	**99.02**	**99.09**	**100%**	**dDDH: 44**	**48**	**47**	**36**	**31**	**32**
												**ANI: 92**	**93**	**93**	**89**	**87**	**87**
***Corallococcus coralloides* T**	**99.67**	**99.61**	**99.09**	**99.74**	**98.83**	**98.7**	**98.83**	**97.66**	**98.63**	**99.74**	**98.96**	**100%**	**dDDH: 44**	**44**	**34**	**30**	**30**
													**ANI: 92**	**92**	**88**	**87**	**87**
***Corallococcus carmarthensis* T**	**99.22**	**99.28**	**99.93**	**99.28**	**99.15**	**99.02**	**99.15**	**98.31**	**98.96**	**99.28**	**99.8**	**99.02**	**100%**	**dDDH: 48**	**36**	**31**	**31**
														**ANI: 93**	**89**	**87**	**87**
***Corallococcus aberystwythensis* T**	**99.35**	**99.15**	**99.8**	**99.15**	**99.15**	**99.02**	**99.15**	**98.31**	**98.96**	**99.26**	**99.67**	**99.02**	**99.87**	**100%**	**dDDH: 35**	**31**	**31**
															**ANI: 89**	**87**	**87**
***Corallococcus* Z5C101001**	**98.76**	**98.83**	**99.48**	**98.83**	**99.22**	**99.22**	**99.22**	**98.11**	**99.15**	**98.83**	**99.61**	**98.57**	**99.54**	**99.41**	**100%**	**dDDH: 32**	**32**
																**ANI: 88**	**88**
***Corallococcus* ZKHCc1_1396**	**98.7**	**98.89**	**98.89**	**98.76**	**99.67**	**99.35**	**99.67**	**97.4**	**99.28**	**98.76**	**99.02**	**98.5**	**98.96**	**98.83**	**99.09**	**100%**	**dDDH: 35**
																	**ANI: 88**
***Corallococcus* CA053C**	**98.57**	**98.5**	**98.7**	**98.5**	**99.54**	**99.67**	**99.54**	**97.59**	**99.61**	**98.63**	**98.83**	**98.37**	**98.76**	**98.76**	**99.22**	**99.28**	**100%**

**Table 4 microorganisms-09-01376-t004:** Differentiation chart comparing SCPEA002 and SCHIC003 draft genome data with sequenced members of the genera *Myxococcus* and *Pyxidicoccus*. The top half uses total genome comparison methods (ANI and dDDH) while the bottom half uses 16S rRNA sequence for pairwise comparison. Orange shading represents species that would be designated as the same using the designated method. Blue shading represents unique species using the designated method, <98.65% 16S identity%, or < 95%/70% for ANI/dDDH.

	SCPEA002	SCHIC003	*M. fulvus* 124B02	*P. fallax* T	*M. stipitatus* T	*M. hansupus*	*M. eversor* T	*M. llanfair* T	*M. vastator* T	*M. virescens* T	*P. trucidator* T	*P. caerfyrddinensis* T	*M. xanthus* DK1622	*M. macrosporus* T
**SCPEA002**	100%	dDDH: 22	23	28	22	23	23	28	25	24	29	34	24	24
		ANI: 82	82	85	82	83	82	82	83	83	86	88	83	83
**SCHIC003**	99.15%	100%	dDDH: 26	23	49	22	27	27	23	22	22	22	22	22
			ANI: 84	82	93	81	85	85	82	82	82	82	81	82
***M. fulvus* 124B02**	99.61%	99.41%	100%	dDDH: 23	26	22	28	28	23	23	23	23	22	23
				ANI: 82	84	82	85	85	82	82	82	82	82	82
***P. fallax* T**	99.54%	98.96%	99.41%	100%	dDDH: 23	24	23	23	25	25	30	29	25	25
					ANI: 82	83	82	82	84	83	86	86	83	84
***M. stipitatus* T**	99.15%	100%	99.41%	98.96%	100%	dDDH: 22	27	27	23	22	22	22	29	22
						ANI: 81	85	85	82	82	82	82	81	82
***M. hansupus***	98.89%	98.44%	98.89%	98.57%	98.44%	100%	dDDH: 22	23	32	32	24	24	31	32
							ANI: 82	82	88	87	83	83	87	88
***M. eversor* T**	98.83%	98.24%	98.83%	98.50%	98.24%	99.15%	100%	dDDH: 41	23	22	24	23	22	23
								ANI: 91	82	82	82	82	82	82
***M. llanfair* T**	98.76%	98.31%	98.89%	98.44%	98.31%	99.09%	99.93%	100%	dDDH: 23	23	24	23	23	23
									ANI: 82	82	83	82	82	82
***M. vastator* T**	98.70%	98.24%	98.70%	98.37%	98.24%	99.41%	98.96%	98.89%	100%	dDDH: 52	25	25	52	41
										ANI: 94	84	84	94	91
***M. virescens* T**	98.63%	98.14%	98.63%	98.31%	98.18%	99.35%	98.89%	98.83%	99.67%	100%	dDDH: 25	24	73	40
											ANI: 83	83	97	90
***P. trucidator* T**	99.09%	98.37%	98.70%	99.02%	98.37%	98.89%	98.83%	98.76%	98.70%	98.63%	100%	dDDH: 33	24	25
												ANI: 88	83	84
***P. caerfyrddinensis* T**	99.48%	98.76%	99.09%	99.41%	98.76%	99.15%	98.96%	98.89%	98.96%	98.96%	99.61%	100%	dDDH: 24	25
													ANI: 83	83
***M. xanthus* DK1622**	98.57%	98.11%	98.57%	98.24%	98.11%	99.28%	98.83%	98.76%	99.74%	99.93%	98.57%	98.83%	100%	dDDH: 40
														ANI: 90
***M. macrosporus* T**	98.89%	98.44%	98.89%	98.57%	98.44%	99.48%	99.15%	99.09%	99.67%	99.61%	98.89%	99.15%	99.54%	100%

**Table 5 microorganisms-09-01376-t005:** Overview of BiG-SCAPE BGC sequence similarity networks of the six strains under investigation in this study.

Myxobacteria	# of Total BGCs	# and % of Singletons	# of Edges Formed with other BGCs	# of BGCs with 1 or 2 Edges	# of BGCs with 3 or More Edges
*A. primigenium* ATCC 29037	32	24 (75%)	21	6	2
*C. macrosporus* ATCC 29039 *	42	9 (21.4%)	228	4	29
NCSPR001	32	1 (3.1%)	248	7	24
NCCRE002 *	36	3 (16.7%)	231	7	26
SCPEA002	34	26 (76.5%)	62	4	4
SCHIC003	29	8 (27.6%)	85	13	8

* genomes with fragmented biosynthetic pathways, likely resulting in fewer clustered pathways than truly exist.

## Data Availability

Sequencing data have been deposited in NCBI under the accession numbers JADWYI000000000.1, JAFIMU000000000, JAFIMS000000000, JAFIMT000000000, CP071090, and CP071091 for strains *A. primigenium*, *C. macrosporus*, NCSPR001, NCCRE002, SCPEA002, and SCHIC003, respectively.

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
