# Peer review of "Assessment of Evolutionary Relationships for Prioritization of Myxobacteria for Natural Product Discovery"

_microorganisms, 2021, doi:10.3390/microorganisms9071376_

Round 1

Reviewer 1 Report

Manuscript ID: microorganisms-1187135

Assessment of Evolutionary Relationships for Prioritization of Myxobacteria for Natural Product Discovery

By: Andrew Ahearne , Hanan Albataineh , Scot E Dowd , D. Cole Stevens

The finding that A. primigenium turns to be a member of the Mellitangium clade is somehow not a surprise to me based on the phylogenomic analysis. Modern taxonomy is the current trend and will stay in the future. However, the classical approach is sometimes helpful to complement and seems important as well to guide initially in the isolation of myxobacteria. It is good that this study clarifies the taxonomic placements of those six strains used in this study and emphasized the taxa needed to be prioritized for the discovery of novel natural products. From what I see, I think the authors should consider to include all the type strains in their phylogenomic analysis to get an overall view of the study and to have a clear species delineation.  Some specific comments and suggestions are stated below.

Specific comments and suggestions:

Page 1. First line in the Introduction section stating “Over the last decade…family Myxococcaceae.” This statement is not correct. Please carefully check the numbers. Myxococcaceae should be in italics.  

Page 2. Supplemental Table1 is not consistent with other supplemental Tables

Page 2. Please indicate superscript “T” in strain DSM 2259 to indicate that it is a type strain. Also, please indicate everywhere in the manuscript for the designated type strain including in all phylogenetic trees.

Page 2, Figure 1 caption. I would suggest changing the word “samples” to strains to be more specific. Also, the caption should be something like “Myxobacterial fruiting bodies from strains NCSPR001, etc.”

Page 2, last paragraph. It would be nice to include some additional information, history, or references about the two ATCC strains used in this study and not just merely the year it was isolated.

Page 3, Isolation of Environmental Myxobacteria. “Detailed location… (Table S1). This is referred to a wrong supplemental.

Page 3, Isolation of Environmental Myxobacteria. - There is missing information about the incubation temperature, time, and other details in the isolation of myxobacteria.

Page 3, “To prepare the baiting plate… (TSB) with agar” – What agar percentage?

Page 4, BIG-SCAPE analysis - Please indicate the strains and their genome accessions used in this analysis, or cite the reference.

Page 4, First line: Cystobacterineae should be in italics.

Page 4, AntiSMASH – Please indicate the “Detection strictness setting” used in the analysis.

Page 4, Results section. “Overall, the assemblies…myxobacterial genomes.” Which genomes are referred to as “recently sequenced myxobacterial genomes”? Maybe a reference is needed in this line.

Page 5, Figure 2. I would suggest to include all type strain genomes in this phylogenetic tree to see better the relationship. Also, please mark the type strains. It would also be nice to see a 16S rDNA phylogenetic tree that would include the six strains (4 isolates, 2 ATCC) in this study.

Page 6, Table2. “S” in 16s should be capitalized.

Page 7, Figure 3. This should be a Table and not a Figure.

Page 8, Figure 4. This is a Table and not a Figure.

Page 9, Figure 5. Cystobacterineae should be in italics.

Page 10, Table 3. For consistency, would be nice if the ATCC numbers are also indicated for A. primigenium and C. macrosporus.

Supplementary Data:

Table S1. Please indicate the strain number and their genome accession.

Table S2. This is wrongly cited in the manuscript. See earlier comment (Page 3, Isolation of Environmental Myxobacteria).

Figure S1, S2. Indicate the type strains.

Figure S3. Please included all the type strain genomes in this tree and indicate the type strains.

Author Response

*Please see the attachment.

Reviewer 2 Report

This paper is an interesting study but I am a bit confused about the aims.

The authors report various new strains that may represent novel taxa but they do not make any attempts to formally describe them. I can not really see a reason for this "strategy" except that I suspect that they will want to make formal descriptions in another journal where this is possible. But if this is the case, the authors should emphasize more on the second part of the manuscript, i.e. the genome mining for BGC encoding for secondary metabolites where the data are still a bit vague.

There is also some confusion about the taxonomic concepts and this should be streamlined.

The genome analysis is generally well done and innovative but in particular because the secondary metabolites and their biosynthesis have been highligted I would have expected that the authors show the chemical structures of these compounds and explicitly search for the individuel BGC of further metabolites that are known from these bacteria. The coralloypronin BGC is such a rather striking example. The authors only cite the genome announcement of C. coralloides but have not mentioned the recent paper on the biosynthesis https://www.sciencedirect.com/science/article/abs/pii/S1096717619300266

A similar case is the disciformycin from P. fallax.

The presentation of the results in this context could greatly benefit from the inclusion of such data and also a more elaborate treatment of the known secondary metabolites from the mentioned genera to which these new species belong. 

I made several comments in the pdf and hope those will be helpful.

The reference section is a mess. The citations are all in incorrect format and this must by all means be improved, in particulart with regard to font and capitalization. I have only highlighted some examples.

Overall I think that this paper can be published, but some major revisions are necessary before acceptance can be envisaged.

Author Response

*Please see the attachment.

Reviewer 3 Report

The manuscript by Ahearne et al looks at the evolutionary relationships of various “natural” products from several Myxobacteria using various genomic analyses and comparisons.  Overal this is nice piece of work, but somewhat incomplete. 
First, the authors did not look at a variety of other genomic and evolutionary papers focused on the myxobacteria.  Several published works by G. Sharma are missing.  While his work does not focus on natural products, it focuses on motility and pili, as well as various taxis systems the evolutionary concepts he suggests should be examined and compared to the authors predictions.  It should also be noted that G. Sharma examined 34 species in 2018, I would assume there were several more added or at least partials added since then.  In addition Table S1 should include references as to who described them, in addition to the year and NCBI:txid.  Please double check item 2 in the table, “Myxococcus   llanfairpwllgwyngyllgogerychwyrndrobwllllantysiliogogogochensis sp. nov.” seems like a typo.

Second, this reviewer would like to see a more robust analysis of the novel isolated strains against all other myxobacteria genomes identified.  Much of the data seems to be focused on just a few species closly related.  A broader analysis may provide information lost by limiting the comparisons.  For example, Table 1 only compares the unknowns to 2 known myxobacteria.

My third issue with the manuscript is the figures, Figures 3, 4, and 5 in particular.  The descriptions in the legends are nonexistent.  This reviewer was confused by the multiple copies of the some of the figures, figures also labeled Tables etc.  Not sure what pdf I downloaded ( peer-review v2) but I could not tell what belonged to what.  Please provide detailed legends so the reader can follow along and understand the figures and tables.  Figure 1 should be marked with labels, A-E along with the strain name.  Why do they look purple? Is SCHIC003 really purple?  I would suggest a reshoot of the colonies.  I would also suggest pictures of the fruiting bodies.   Table 2 heading is incorrect, M. xanthus is not included and that is certainly a fully-sequenced myxobacteria.  There are more than 5 fully sequenced myxobacteria.  This is misleading as written.   

Finally, the work is incomplete and the authors need to broaden their view of comparisons for these various Natural Products.  In addition, they need to look closer at the work of others doing Myxobacterial evolution studies.

Round 2

Reviewer 1 Report

It was frustrating that the comments were not taken seriously. From what I see in the revised version of Figure 2, Figure S1, Figure S4, Table 3, Table 4, and Table S4, there’s no consistency for indicating the type strains, and NOT ALL type strains were considered. Am not sure if the authors can identify the correct type strain. Also, one of the biggest concerns is not including ALL the type strains with genomes. This can be seen for example in Figure 2 and Figure S4. In the 10 genome-sequenced of Corallococcus type strains, only 3 were included while only two (out of eight) for Myxococcus type strains (referring to Figure 2). This is also the same case in Figure S4 where only 4 (out of eight) type strains were included. Delineating the genomic and evolutionary relationship is hard to prove and impossible without consideration of all the type strains.

Minor comment:

Reference 49 seems not referring to the two ATCC strains and can be deleted.

Reviewer 2 Report

I have seen that my concerns have been addressed and recommend that this paper should now be accepted

Reviewer 3 Report

I am disappointed that the authors did not take all of my suggestions as seriously as I would have liked.  In particular the rational for the strategy was dealt with.  The authors should make this manuscript high priority and include all data to support it, not wait for additional possible manuscripts. 

Second the discussions of the analysis and the discussion of a broader more scholarly analysis with other myxo evolution papers is lacking.   More robust analysis and data would greatly help the quality of the paper.